# Structural Insights into RNA Dimerization: Motifs, Interfaces and Functions

**DOI:** 10.3390/molecules25122881

**Published:** 2020-06-23

**Authors:** Charles Bou-Nader, Jinwei Zhang

**Affiliations:** Laboratory of Molecular Biology, National Institute of Diabetes and Digestive and Kidney Diseases, 50 South Drive, Bethesda, MD 20892, USA

**Keywords:** RNA, intermolecular interaction, dimerization, structure, domain swapping, folding, ribozymes, riboswitches

## Abstract

In comparison with the pervasive use of protein dimers and multimers in all domains of life, functional RNA oligomers have so far rarely been observed in nature. Their diminished occurrence contrasts starkly with the robust intrinsic potential of RNA to multimerize through long-range base-pairing (“kissing”) interactions, self-annealing of palindromic or complementary sequences, and stable tertiary contact motifs, such as the GNRA tetraloop-receptors. To explore the general mechanics of RNA dimerization, we performed a meta-analysis of a collection of exemplary RNA homodimer structures consisting of viral genomic elements, ribozymes, riboswitches, etc., encompassing both functional and fortuitous dimers. Globally, we found that domain-swapped dimers and antiparallel, head-to-tail arrangements are predominant architectural themes. Locally, we observed that the same structural motifs, interfaces and forces that enable tertiary RNA folding also drive their higher-order assemblies. These feature prominently long-range kissing loops, pseudoknots, reciprocal base intercalations and A-minor interactions. We postulate that the scarcity of functional RNA multimers and limited diversity in multimerization motifs may reflect evolutionary constraints imposed by host antiviral immune surveillance and stress sensing. A deepening mechanistic understanding of RNA multimerization is expected to facilitate investigations into RNA and RNP assemblies, condensates, and granules and enable their potential therapeutical targeting.

## 1. Introduction

Quaternary structures of biological macromolecules are frequently at the core of functional assemblies that enable life. Such higher-order structures form through a network of intermolecular interactions between individual modules employing recurring interfacial motifs to drive multimerization. Association of identical or different building blocks produce homo or hetero-multimers, respectively. These oligomeric assemblies often lead to overall structural symmetry or pseudosymmetry that can be important for biological function and evolution [1,2,3,4]. More than 65% of proteins are believed to rely on homo-oligomeric states for their activities [1,5]. By comparison, only a handful of naturally occurring RNAs are known to function in homodimeric states, despite the frequent occurrences of palindromic or complementary, dimer-forming sequences and a plethora of RNA structural motifs that can engage specific and robust long-range contacts [6]. One of the most striking examples of a functional RNA dimer lies in the peptidyl-transferase center (PTC) of the ribosome. In the PTC, a cleft formed between a pair of near-symmetrical RNA helical motifs is proposed to catalyze ancient chemical reactions, such as peptide bond formation, constituting a dimeric proto-ribosome ribozyme [7].

The simplest, most common form of RNA-RNA interactions occur through base pairing of complementary single-stranded sequences, forming heterodimeric double-stranded RNA (dsRNA). These specific interactions are widespread in diverse forms of post-transcriptional gene regulation and RNA modification and processing, such as the pairing of small interfering RNAs (siRNA), microRNAs (miRNA), or small nucleolar RNAs (snoRNAs), etc., to their complementary RNA targets. [8,9,10,11,12]. These hybridization events produce heterodimeric dsRNAs that are abundant in cells. However, for the purpose of this review, we generally exclude simple dsRNAs that form solely through sense-antisense hybridization and treat dsRNAs as monomers to focus on dimerization that involve tertiary structural contacts.

Consecutive base pairing positions adjacent base planes to stack with each other, either within the same strand or crossing into the opposing strand—termed cross-strand stacking. Notably, stacking can also occur without base pairing as the aromatic stacking itself provides significant favorable enthalpy. These non-covalent, attractive interactions between polarized aromatic nucleobases daisy-chain the base pairs into helices and frequently further concatenate adjacent helices to form long, coaxial helical stacks. In so doing, aromatic-aromatic (or π-π) interactions to a large extent dictate the overall shape of most RNA tertiary and higher-order structures. In addition to base pairing and stacking, other types of tertiary interactions also contribute to RNA-RNA interactions, including dimerization, such as purine-minor groove interactions, ribose zippers, tetraloop-tetraloop receptor interactions, etc. These tertiary and quaternary interactions between discrete RNA elements can be dramatically stabilized by the presence of Mg^2+^ ions and, to a lesser extent, by monovalent cations, such as K^+^. Mg^2+^ ions play the dual roles of serving as diffuse counter ions that ameliorate the electrostatic stress from juxtaposing densely charged phosphate backbones, as well as bridging specific tertiary contacts in the form of chelated ions [13].

In contrast to their apparent biological rarity in cells, RNA multimers frequently emerge in vitro and often present a nuisance in the laboratory. Most of such in vitro multimers are believed to occur via fortuitous pairing of short segments of complementary sequences, when the RNA was heat-denatured and cooled slowly—a process that encourages strand annealing. As such these in vitro multimers are similar in origin to those dsRNAs produced by sense-antisense hybridization events in cells. Various RNA refolding procedures have been developed with an explicit or implicit objective to enrich monomers, while dimers and higher oligomeric states are reduced and usually discarded as artifacts or physiologically irrelevant entities. For instance, many unmodified tRNAs exhibit strong tendencies to dimerize and oligomerize and require a “snap-cool” procedure to refold into monomers [14]. However, the view that RNA oligomers are largely refolding artifacts is rapidly changing due to the recent recognition that multivalent, intermolecular RNA-RNA contacts can induce functional RNA oligomerization, condensation, phase separation or formation of ribonucleoprotein (RNP) granules that critically regulate cellular metabolism and cell fate [8,15,16,17,18,19]. These recent findings accentuate the importance and previously underappreciated role of RNA multimerization in biology. Although our molecular understanding of RNA quaternary structures is still in its infancy, RNA nanotechnologies have utilized artificially engineered RNA assemblies for more than two decades to create programmable supramolecular architectures [20,21,22,23,24,25].

In order to understand how RNA dimerizes and multimerizes in general, we survey a representative collection of RNAs that rely on their homodimeric states to exert their biological functions and also those that dimerize fortuitously, such as in crystallo. Interestingly, in several riboswitches, we are able to compare the monomeric and dimeric forms of essentially the same RNA. We maintain a focus on RNA dimerization interfaces and structural motifs used to drive self-assembly with an emphasis on tertiary contacts in addition to base pairing. This meta-analysis revealed that kissing-loop interactions and complementary strand swapping are the predominant dimerization motifs. Since only a handful of recurring RNA structural motifs are responsible for mediating dimerization in most instances, we organize the discussions based on RNA functional classes instead of the dimerization motifs that they employ. In the following sections, we examine instances of RNA dimerization that occur in retroviral genomes, mRNA localization, ribozymes and riboswitches, and fluorogenic RNAs, as well as consequences of RNA multimerization in diseases and immunity.

## 2. Dimerization of Retroviral Genomic RNAs 

Perhaps the most studied physiologically relevant RNA homodimerization phenomenon is the non-covalent linkage of two viral genomic RNA (gRNA) copies in retroviruses [26,27,28,29,30,31]. This conserved quaternary architecture of several kilobases of gRNA enable multiple biological functions in the viral lifecycle. First, templated genomic repair during reverse transcription (RT) is facilitated by switching of the reverse transcriptase between the two spatially close strands [32]. Second, genetic diversity is enhanced by homologous recombination between the two gRNA copies [33]. Lastly, interconversion between the monomeric and dimeric gRNAs modulate their translation and packaging into viral particles [34,35]. The latter is controlled by modulating the exposure of binding sites for viral nucleocapsid (NC) domains [30,36,37]. Retroviral gRNA dimerization frequently occurs through a segment of their 5′ leader region termed the dimer linkage site (DLS), as detailed below [38,39,40] (Figure 1).

### 2.1. HIV-1 gRNA

In the case of human immunodeficiency virus type-1 (HIV-1), a dimerization initiation site (DIS) was identified within the ~ 350 nucleotide DLS [41,42]. The DIS contains a six-nucleotide GC-rich palindrome of the sequence GUGCAC in an apical loop. Mutation of this palindromic sequence blocks gRNA dimerization, while compensatory mutations restore it. Importantly, the DIS can form two types of dimers in vitro that have been extensively characterized structurally. A kissing loop dimer is formed at low temperatures in the presence of Mg^2+^ [43,44,45,46,47] (Figure 1b,c), while a more stable extended duplex dimer forms through refolding at higher temperatures or in the presence of NC acting as a chaperone [48,49,50,51,52] (Figure 1d,e,f). The kissing loop dimer is considered a transient intermediate leading to the extended dimer. Curiously, dimeric gRNAs extracted from mature virions are more stable than those from immature viral particles [53]. Thus, it is possible that a transition from a loose kissing loop dimer to a more stable extended DIS occurs during viral maturation.

The crystal and nuclear magnetic resonance (NMR) structures (Figure 1b,c) of the DIS kissing loop show an identical self-pairing of the palindromic GUGCAC loop [43,44]. The flanking 3′ A (A16 in Figure 1b,c) stacks on one of the closing Watson-Crick pairs. The DIS palindromic sequence is flanked by strictly conserved purines whose substitution or deletion block dimerization [54,55]. Thus, this cross-strand stacking is required to stabilize the dimeric interface. Nonetheless, the crystal structure shows a bulged-out conformation of the flanking 5′ purine, while the NMR structure reveals a non-bulged conformation. This hints at the flexibility of the DIS kissing loop which could be required to transition towards a more stable extended duplex, a notion supported by single-molecule FRET analysis [56]. Structures of the extended dimeric DIS reveal an A-form dsRNA structure generated by self-pairing of the palindromic loop and strand swapping between 5′ and 3′ strands of both protomers (Figure 1d,e,f) [48,49,52]. This more than doubles the surface of the dimerization interface (Table 1, compare PDB 1BAU & 2B8S versus 1Y99 which have similar sequence lengths) rationalizing the increased stability. In the extended DIS crystal structure, the RNA is colinear with a single bulged out nucleotide. In contrast, the NMR and cryo-electron microscopy (cryo-EM) structures reveal bent conformers of the extended DIS. In the latter, an S-turn at the 5′ region was observed but it remains unknown if this RNA motif only forms in the context of the extended DIS duplex.

Conformational changes in the HIV-1 5′ leader regulate the oligomeric state of the gRNA [52,57,58]. The palindromic loop of the DIS is sequestrated intramolecularly by pairing to the U5 element in the monomeric DLS [59,60,61] (Figure 1a). Exposure of the DIS is achieved by pairing of the Gag start site (AUG in Figure 1a) to the U5 thus inducing the dimeric DLS required for genome packaging. AUG was proposed to pair either in *cis* or in *trans* with the U5 element of the same or the other gRNA copy. It is tempting to speculate that the differences in stability of dimeric gRNAs between immature and mature viral particles could originate from the nature of AUG pairing to U5 (*cis* vs. *trans*). The structure of a minimal HIV-1 RNA packaging signal in its monomeric state consisting of the U5, AUG, DIS mutant and splice donor (SD) was recently determined [62,63]. Further work is needed to fully describe the dimerization interface of the HIV-1 DLS and decrypt the monomer to dimer structural switch [35]. Indeed, the adjacent trans-activation response element (TAR) has also been proposed to directly or indirectly contribute to gRNA dimerization, as TAR mutations and deletions exhibited strong influences on gRNA dimerization [64,65,66,67]. Furthermore, tRNA^Lys3^ annealing to the primer binding site (PBS) in the DLS was suggested to enhance gRNA dimerization [61,68]. Taken together, it is likely that multiple structural elements (DIS, AUG, U5, TAR, PBS, and even poly-A) within the dynamic 5′ leader coordinately modulate the exposure of the DIS palindrome, nucleation of the gRNA monomers and subsequent structural re-organizations enabling propagation of the dimer interface. Conceivably, other RNA viral genomes may utilize similar strategies to sequentially engage multivalent contacts, as well as morph global and local structures, to ultimately achieve prescribed dimer configurations.

### 2.2. MoMuLV and MoMuSV gRNAs

Another well-characterized retroviral dimeric gRNA is the ~380 nucleotide DLS of Moloney murine leukemia virus (MoMuLV) and the closely related Moloney murine sarcoma virus (MoMuSV) [70,71,72,73,74]. Switching between a monomeric and dimeric gRNA is achieved by four dimerization motifs. Intermolecular pairing of two distinct palindromic sequences PAL1 and PAL2 induce a register shift in the DLS which exposes self-complementary GACG loops in SL1 and SL2 (also named SL-C and SL-D) (Figure 1g). Remarkably, the NMR structure of the isolated SL2 from MoMuSV reveals a stable homodimer mediated by a kissing loop [73] (Figure 1h, PDB 1F5U). This is among the smallest RNA dimeric interface reported to date spanning 247 Å^2^ (Table 1). Only two base pairs are formed between both palindromic loops (C10:G11′ and G11:C10′, Figure 1h) and an adenosine stacks on each side of the kissing loop helix (A9 and A9′ in Figure 1h). The NMR structure of the MoMuLV domain SL1-SL2 shows an identical dimeric interface as the isolated MoMuSV SL2, but formed between the loops of SL1 and SL2 with SL2′ and SL1′ of the second monomer, respectively [74] (Figure 1g,i). It is likely that the strength and bend of the stems act as topological tools to restrict the intermolecular self-pairing to SL1-SL2′/SL2-SL1′ instead of SL1-SL1′/SL2-SL2′. 

### 2.3. Other Viruses

Several other classes of viruses also rely on RNA dimerization for specific functions. A palindromic kissing loop in the SARS coronavirus genome was shown to regulate ribosomal frameshifting, thus affecting viral growth [75]. The 3′ untranslated region (UTR) of the hepatitis C virus can form two alternative dimeric interfaces [76,77,78,79]. One relies on the complementarity between the DLS in the 3′-UTR and the distal stem-loop 5BSL3.2 of the 5′-UTR which is proposed to enhance translation. The other dimer is mediated by a palindromic kissing loop in the 3′-UTR and proposed to regulate genome packaging.

In summary, a number of RNA viruses, including retroviruses, rely on kissing loop-loop interactions through palindromic or other complementary sequences to induce genome homodimerization for various essential functions. To buttress these generally short duplexes, single-stranded purines frequently cross-strand stack with both flanks of the intermolecular helices stabilizing them. The stacking of such bases contributes significantly to the stability of the dimeric interface by minimizing exposure of hydrophobic bases to the solvent and by favorable enthalpy changes through the aromatic interactions [80]. This strategy is reminiscent of the stabilization of codon-anticodon interactions in the ribosome [81,82] and T-box riboswitches [83,84,85,86], as well as a 3-bp pseudoknot in the adenovirus VA-I RNA [87].

## 3. Dimerization-Mediated mRNA Localization

The 3′-UTRs of eukaryotic mRNAs are regulatory hubs for their translation, decay and cellular localization, etc. [88]. Non-uniform distributions of certain mRNAs in the cell create spatio-temporal patterns of gene expression and localized translation which are crucial for cell polarity during cell differentiation and embryo development, etc. 3′-UTRs contain sequence and structural elements that can recruit distinct protein partners to exert localized functions [89]. Some of these 3′-UTR signatures act as “zip-codes” and dictate the cellular localization of the mRNA. Diverse mechanisms orchestrate mRNA localization and have been extensively reviewed [90,91,92,93,94,95]. A peculiar case is the targeting of the *oskar* (*osk*) and *bicoid* (*bcd*) mRNAs to opposite poles of *Drosophila* embryos [96,97,98]. In both cases, the Staufen protein [99,100] mediates the transport of both mRNAs through movement on microtubules, while other factors repress translation until *osk* or *bcd* reach their respective destinations [101,102,103]. Importantly, homodimerization of these mRNAs is required for efficient subcellular localization.

### 3.1. Oskar mRNA

Spliced *osk* is localization-competent, while unspliced *osk* is not [104]. Nonetheless, the latter can be localized at the pole of the cell by hitchhiking onto spliced *osk* through intermolecular interactions. A conserved six-nucleotide, GC-rich palindromic loop sequence in the region of 714-827 of the *osk* 3′-UTR was shown to promote homodimerization in vitro [105] (Figure 2a). Such a kissing loop-loop interaction is reminiscent of the DIS of HIV-1 (see above). However, the conversion into an extended duplex has not yet been observed for *osk* and may not form due to a larger loop flanked by pyrimidines. Double nucleotide substitutions in the palindromic sequence blocked dimerization by preventing kissing interactions. Importantly, dimerization is not required for localization of spliced *osk* but necessary for hitchhiking by unspliced *osk* [105]. Although the functional role of *osk* localization or translation activation through dimerization remains unclear, it exemplifies how mRNAs lacking “zip-codes” could still be targeted to specific cellular regions by piggybacking with other RNAs through RNA multimerization.

### 3.2. Bicoid mRNA

The localization signal of *bcd* mRNA, found in the domain III of its 3′-UTR, was shown to form dimers in vitro [106,107,108]. Pairing of the apical loop of domain III with a complementary internal bulge sequence form the dimerization motif via six Watson-Crick pairs (Figure 2b). At least two sequential steps are involved during self-assembly and are kinetically controlled [107]. Open dimers are formed initially through pairing of a single motif between two protomers. This initial nucleation event then leads to a closed and stable (nearly irreversible) dimer with both motifs paired (Figure 2b). Mutations that disrupt the complementarity between the loop and the bulge block dimerization in vitro and inhibit cellular targeting of *bcd* in vivo [106,107]. This suggests that *bcd* multimerization could bring into close proximity different duplex regions of the 3′-UTR, which in turn recruit one or multiple double-stranded RNA-binding domains (dsRBDs) [109] of Staufen to form large ribonucleoprotein particles prior to loading on microtubules.

Interestingly, *bcd* forms exclusively dimers in vitro, while the isolated domain III assembles into dimers, trimers, and tetramers [107]. It is likely that neighboring regions of domain III sterically and topologically affect its oligomerization. Furthermore, protein factors may also impact *bcd* quaternary structure. This multimerization mechanism and the ability to form more than one type of multimer is reminiscent of the bacteriophage ø29 prohead RNA (pRNA), discussed below [110,111,112,113] (Figure 2c).

### 3.3. ø29 Prohead RNA

The bacteriophage ø29 pRNA is a component of the packaging motor required for viral genomic DNA encapsulation. pRNA forms dimers in vitro through a kissing loop interaction mediated by four complementary base pairs between two distinct stem-loops [114,115]. However, pentamers or hexamers of pRNA are thought to be the functional assemblies and form through interactions with other protein factors of the packaging motor (protein connector, protein capsid, and ATPase). In this model, pRNA “open homodimerization” is proposed to be the nucleation point that precedes the formation of higher-order oligomers. Structures of the pentameric and tetrameric pRNAs [111,112,113] (Figure 2c) revealed that a range of oligomeric assembly configurations can be produced by mix-and-matching two complementary loop sequences on either end of the RNA protomer in head-to-tail configurations. The intrinsic structural flexibility of the RNA protomer may permit formation of a linear or circular chain of protomers of variable lengths. This use of two complementary but non-palindromic sequences to build multimers contrasts with the tendency of RNAs bearing single palindromic sequences to form dimers. This architectural modularity has been extensively used in RNA nanotechnology [21,23,116,117,118].

## 4. Homodimeric Ribozymes

Ribozymes are RNA catalysts that accelerate various chemical transformations, including self-cleavage, splicing, tRNA aminoacylation and peptidyl transfer, etc. Many naturally occurring ribozymes catalyze post-transcriptional RNA processing, such as backbone cleavage or ligation, and usually proceed through a general acid-base mechanism and produce 2′,3′-cyclic phosphate and 5′ hydroxyl termini [119,120]. Several ribozymes are known to function as homodimers.

### 4.1. VS Ribozyme

The Varkud Satellite (VS) ribozyme, the largest known nucleolytic ribozyme, was discovered 30 years ago in the mitochondria of *Neurospora* fungi. It has both self-cleavage and ligation properties and functions in the processing of RNA intermediates of rolling-circle replication [121]. It is organized into seven helical regions that can be *trans*-cleaved following self-assembly into a homodimer [122,123]. Its structure and mechanism have been the subject of extensive studies [124]. The crystal structure of the entire VS ribozyme revealed an intertwined symmetric dimer spanning an interface of 2612 Å^2^, the largest RNA homodimerization interface described to date (Table 1). Stem-loop 1 (P1) forms a kissing loop interaction with stem-loop 5′ (P5′) of the second protomer through three Watson-Crick base pairs and one non-canonical C629-A696′ pair (Figure 3a) [123,125]. Interestingly, cross-strand stacking by G633 bolsters the 4-bp intermolecular duplex, analogous to the aforementioned retroviral RNAs. This leads to a domain-swapped configuration where helix P1 docks with helices P2′, P5′ and P6′ of the other subunit, thus forming a composite active site. Here, the general base G638 of helix P1 and the general acid A756′ of helix P6′ flank the scissile phosphate between G620 and A621, driving its in-line nucleophilic attack, stabilization of the transition state and ultimately departure of the leaving group. The intermolecular kissing interaction between the P1 and P5′ loops probably initiates VS ribozyme dimerization. Subsequently, another interaction between the middle section of P1 and P2′ and P6′ brings into proximity the reactants forming the catalytic site (Figure 3a). Intriguingly, this second interaction is primarily stacking in nature with the bases of A751′ and C750′ inserted between the aromatic rings of A621 and A639. Despite not employing any base-pairing interactions, this secondary contact likely contributes significantly to dimerization by assembling a coaxial array of 5 nucleobases stabilized by consecutive π-π stacking interactions. 

### 4.2. Hatchet Ribozyme

The Hatchet ribozyme is a small ribozyme that was recently discovered in several genes of *Veillonella sp*. [126] and shown to form stable homodimers [127]. The dimerization is driven by a tetranucleotide palindromic internal loop sequence ACGU (nts 67-70) that forms a symmetric helix (Figure 3b). This leads to the exchange of the 3′ strand between protomers forming a hybrid helix P2 (pairing between A21 to G29 with A75′ to U81′) and extending the dimeric interface. Although the palindromic sequence is essential for efficient dimerization, it is not required for catalysis and a monomeric hatchet ribozyme maintained efficient cleavage [127]. Interestingly, two conformers of the homodimeric hatchet ribozyme were trapped in different crystal lattices. Their cores are nearly identical with an RMSD of 1.8 Å over 64 residues but the orientation of the swapped 3′ strands are rotated by ~80° (Figure 3b). This led to two distinct configurations with the protomers either forming an antiparallel *trans*-like symmetric dimer or a more intertwined *cis*-like pseudosymmetric dimer. In solution, both conformers and potentially others are likely sampled as a result of the apparent flexibility of the 3′ extended tail with the embedded palindromic sequence. An unanswered question is whether this dimerization occurs in vivo and for what purpose, since the palindromic sequence is found in the Hatchet ribozymes of multiple organisms but not strictly conserved [126]. 

## 5. Homodimerization of Riboswitches

Riboswitches are *cis*-acting noncoding elements found in the 5′-UTR of some mRNAs (mostly in prokaryotes) [128,129,130]. They regulate the transcription, translation or splicing of downstream genes through direct binding of small-molecule metabolites [131] or tRNAs [132,133] to an aptamer domain, which in turn controls a conformational switch in the expression platform. Most riboswitches function with a single aptamer but some are organized into a tandem arrangement of two to three aptamers that bind the same or even different metabolites, thus linking two or more metabolic pathways [134]. In the following section, we discuss 7 classes of riboswitches that were crystallized as dimers. Among them, 3 classes (Glycine, ZTP, and THF) also crystallized as monomers, allowing direct comparisons with their dimeric counterparts. The biological relevance and potential functions of these dimers will also be discussed. In cases where the tandem aptamers are highly homologous, such as the glycine and guanidine-II riboswitches, dimeric structures of single aptamers in crystallo likely represented the way the natural tandem aptamers interact with each other and act in concert. 

### 5.1. Glycine Riboswitches

One prominent example of tandem riboswitches is the tandem glycine riboswitch formed by two covalently linked glycine aptamers (apt1 and apt 2) followed by a single expression platform [135]. Although it was proposed early on that cooperative binding of individual glycine molecules to each aptamer produced a sharper digital response [135,136,137], later work suggested that each aptamer largely binds glycine independently and the double-aptamer quaternary structure is stabilized by an intervening P0 helix and a K-turn connecting the two aptamers [138,139,140,141]. The crystal structure of an isolated *Vibrio cholerae* glycine apt2 revealed a crystallographic homodimer formed between helix P3 of one protomer and helix P1′ of the other [142] (Figure 4a). Four A-minor interactions [143] stabilize this interface. A40 and A64 from P3 form Type-I A-minor interactions with G84′-C5′ and G7′-C82′ from P1′, respectively, while both A41 and A63 engage Type-II A-minor interactions with the same U6′-A83′ pair (Figure 4a). In addition, extrusion and intermolecular stacking between A65 and A65′ stabilize the crystallographic homodimer and facilitates ligand binding by flipping A65 outside of the glycine-binding pocket. Remarkably, the manner by which this apt2-apt2′ homodimer forms in crystallo, initially thought to be physiologically non-functional, is nearly the same as the way a natural tandem *Fusobacterium nucleatum* glycine riboswitch forms an apt1-apt2 intramolecular interface (RMSD of 2.66 Å over 104 residues; Figure 4a). This is driven by structural similarity between the two aptamers and inherent symmetry of the dimer interface [144].

Further studies showed that singlet glycine riboswitches bind glycine with comparable affinities as the more common tandem version, but require a neighboring “ghost aptamer”, which is likely a degenerate, reduced version of the original partner aptamer that maintained the inter-aptamer contacts for structural stabilization. There are multiple proposals on why most glycine riboswitches have retained a tandem arrangement, as such an arrangement does not apparently enhance ligand binding (at least in vitro) [145,146,147,148,149,150]. Here, we describe one additional hypothesis. The ancestral glycine binding aptamer, presumably “A”-shaped like their descendent aptamers, may have by chance folded into a structure that is in a sense “palindromic”. That means the donors (A-rich bulge or loop) and receptors (minor groove of G-C/C-G pairs) of the A-minor interactions were spaced and angled in such a way that two oppositely oriented copies of the aptamer can simultaneously engage two sets of A-minor interactions with each other, thereby seeding the dimer configuration. This notion is analogous to the design of a dimer tectoRNA, which took advantage of appropriately spaced tetraloop and tetraloop receptor motifs, achieving a dimerization *K*_d_ of ~4 nM [151,152,153]. The resulting mutual reinforcement of both glycine-binding aptamers, clearly advantageous for folding, overall stability, and possibly also tighter binding of small ligands, such as glycine, led to retention of the dimeric form. Through evolution, depending on gene-specific regulatory needs, certain tandem glycine riboswitches had degraded into singletons that function with a ghost aptamer [146]. For tandem adenosines embedded in a helix to effectively engage the minor groove, the donor helix needs to be inclined or angled relative to the recipient helix [143]. This incline presumably produced the ~60° angle between the two connecting helices (P1 and P1′) at the inter-aptamer junction, which then drove the evolution of a K-turn. Once adopted, the stable K-turn could have taken charge of the overall dimer architecture, playing a key role in placing donor adenosines in close proximity to their receptor minor grooves. The A-minor interactions, compared to kissing loops and other base-pairing contacts, lack the specificity to find each other and engage unassisted. Thus, this theory rationalizes the critical requirement of the K-turn and the appearance of ligand-binding cooperativity in its absence. A similar scenario was recently observed in the T-box riboswitches, where an essential K-turn brings the Stem II S-turn motif to dock, via an extended ribose zipper, with Stem I, forming a composite binding groove for the incoming tRNA anticodon [84,85]. Taken together, recent studies of tandem and singlet glycine riboswitches have produced a wealth of important insights not just into ligand recognition by RNA, but into RNA-RNA interactions and the evolution of quaternary structure and geometric motifs. Undoubtedly, the saga of the glycine riboswitches will continue.

### 5.2. ZTP Riboswitches

Two crystal structures of the ZTP (5-aminoimidazole-4-carboxamideriboside 5′-triphosphate) riboswitch were solved (Figure 4b) [154,155,156]. In the *Fusobacterium ulcerans* ZTP riboswitch, two subdomains formed by P1-P2 and P3, respectively, bind each other forming a P4 pseudoknot. This *cis* tertiary interface forms the binding pocket for ZTP. Remarkably, in the *Schaalia odontolytica* ZTP riboswitch structure, this pseudoknot is formed in *trans* by domain swapping in a crystallographic dimer leading to G12-G15 pairing with C54′-C57′ (Figure 4b). The two structures are similar with an overall RMSD of 2.85 Å over 40 residues. Importantly, nearly identical ZTP-binding pockets were observed in the two structures, suggesting that the crystallographic dimer structure from *S. odontolytica* actually captured the biological relevant RNA-ligand interface.

### 5.3. THF Riboswitches

The structure of the *Eubacterium siraeum* tetrahydrofolate (THF) riboswitch exhibited an unraveled P1 helix that strand-exchanged with the P1′ of another monomer forming a scissors-like structure where the intertwined P1-P1′ forms the hinge (Figure 4c) [157]. This homodimerization occurred through seven Watson-Crick pairs between the 5′ G1′-A7′ of one monomer and the 3′ G95-C101 of another (Figure 4c). In addition, C89 forms a Watson-Crick pair with G8′ that is stacked between A10 and G40, while U9 stacks with U9′ further stabilizing this dimeric interface. By contrast, the P1 helix in the *Streptococcus mutans* THF riboswitch remained paired forming a monomer. Both dimeric and monomeric assemblies are structurally similar (Figure 4c), but the latter captured two THF molecules rather than one in the former structure [158]. Both structures brought valuable insights into the function of THF riboswitch and led to the design of artificial homodimeric RNAs responsive to THF through a loop-receptor intermolecular interface [159]. Similar artificial constructs were previously formed through a GAAA tetraloop-receptor mediated homodimerization interface [151,152,160].

### 5.4. Guanidine-II Riboswitches

The guanidine-II riboswitch, originally termed the mini-*ykkC* motif, is the simplest of the three known classes of guanidine riboswitches [161]. It is comprised of two small stem loops (P1 and P2) linked by a variable size linker (7 to 40 nucleotides) [162]. Both stem loops were shown to bind a single guanidine each in a cooperative manner. The ACGR sequence of both loops are highly conserved, shared between both P1 and P2 and were protected by guanidine in in-line probing experiments, while the linker was not. The crystal structures of the individual *Escherichia coli* and *Pseudomonas aeruginosa* P1 and P2 aptamers bound to guanidine revealed homodimeric interfaces through a kissing loop interaction [163,164] (Figure 5a). In both instances, complementary Watson-Crick pairs are formed between C10-G11′ and G11-C10′ from the loops of both protomers, which are further coaxially stacked. This is reminiscent of the kissing loop between SL1 and SL2 of the MoMuSV and MoMuLV discussed previously (Figure 1g–i). Two guanidine molecules bind inside both ACGR loops at the dimerization interface but interact only with one of the protomer. Although this homodimerization occurred in crystallo, it is likely that in the natural full-length guanidine-II riboswitch, a similar kissing loop interaction occurs intramolecularly between P1 and P2. In this context, the dynamics and strength of coaxial stacking might be tuned by the linker for conditional switching. From a thermodynamic perspective, the loop-loop interaction likely pre-organizes the binding site for guanidine, reducing the entropic cost of binding [163,164]. Remarkably, only two base pairs form at the interface. Their engagement is likely facilitated by the covalent P1-P2 tether which brings the pairs into proximity. Further, coaxial stacking across the dimer interface is expected to significantly stabilize the loop-loop interaction and ligand binding. Dimerization presumably requires extension of the variable single-stranded linker, as evidenced by its altered pattern of in-line cleavage upon guanidine binding [162]. It would be interesting to consider how the length and sequence of the P1-P2 linker affect this heterodimerization, ligand binding, and, in turn, gene expression.

### 5.5. Glutamine-II Riboswitches

The recent crystal structure of the *Prochlorococcus sp.* glutamine-II riboswitch has shed light on its ligand-binding properties [165]. The functional form of this gene regulator is likely monomeric but it crystallized as a homodimer (Figure 5b). This occurs through the formation of a 5-bp intermolecular pseudoknot-like helix formed between C1-C5 of the first monomer and G36′-G41′ of the second protomer. Furthermore, a triplex-like interaction involving a base triple (G18•G2-C39′) and the sandwiching of G34′ in between A20 and G22 complete the homodimer interface. Overall, this leads to a domain-swapped dimer configuration with a P1-pseudoknot-P2′ helical stack representing the functional monomeric unit. Notably, a similar domain swapping configuration was seen in the crystal structure of a *Dicistroviridae* internal ribosomal entry site (IRES) [166]. Although these domain-swapped dimer configurations presumably do not occur in vivo, the dimeric structures nonetheless were able to be interpreted to understand the functional monomers.

### 5.6. PreQ1 Class III Riboswitches

Like the glutamine-II riboswitch and *Dicistroviridae* IRES, the *Faecalibacterium prausnitzii* preQ1 class III riboswitch bound to its ligand also crystallized in a homodimeric and partially domain-swapped configuration. In its previously elucidated class I and II counterparts, preQ1 binding at the helical junctions between P1 and P2 stabilizes coaxial stacking and directs helical sequestration of the ribosome-binding site (RBS, or Shine-Dalgarno sequence) [167,168,169,170,171]. Intriguingly, the class III variant has its RBS located more downstream and embedded in a single-stranded 3′ tail region [167]. The homodimer structure revealed that upon preQ1 binding, the 3′ RBS-containing tail formed a robust 6-bp helix with a complementary sequence from the loop of P4 of the other monomer, forming a second pseudoknot [172] (Figure 5c). In this open configuration, U92-C97 of the 3′ tail is paired with G66′-U71′ of the P4′ loop. The resulting helix is further reinforced by cross-strand stacking by G98. Molecular dynamics simulations and single-molecule FRET analysis suggested that the 3′-tail can rapidly dock and undock from the P4 loop within the same monomer (closed configuration), as modulated by preQ1 binding. Although this homodimerization is not known to occur in solution, the same dimerization interface is likely used within a single riboswitch monomer for gene regulation by this novel mechanism.

### 5.7. SAH Riboswitches

The crystal structure of the *Ralstonia solanacearum* S-adenosylhomocysteine (SAH) riboswitch revealed an intermolecular P1 helix formed by strand swapping between two monomers [173] (Figure 5d). Four Watson-Crick pairs formed intermolecularly between U36-G38 of a junctional region (J1/4) and G1′-A3′ of frayed P1 helix of the other monomer. Intriguingly, this created an unusual X-shaped structure that resembles a stacked Holliday junction [174]. Selective 2’-hydroxyl acylation analyzed by primer extension (SHAPE) analysis revealed that the terminal two base pairs of P1 tend to unravel in solution, which likely facilitated strand exchange and the observed dimerization in crystallo [173]. Similarly, this dimerization interface is not thought to function in vivo.

In summary, most known riboswitches carry out their regulatory functions in monomeric states. However, their homodimerization is frequently observed in crystallo through strand or domain swapping. This is likely the result of their conformational flexibility, especially in their gene-regulatory P1 regions that are prone to strand fraying and subsequent selection by the crystallization process. Despite not being completely natural, these dimeric structures have brought essential functional insights into specific recognition of various small molecules by RNA. Intriguingly, most of these crystallographic dimers are structurally similar to their monomeric counterparts, not only in their ligand-binding pockets but also in their overall architectures. This is exemplified by the glycine, ZTP, and THF riboswitches, of which both monomeric and dimeric structures are available for comparison (Figure 4). For riboswitches that are only known to function as singlet monomers but crystallized as dimers, it remains possible that future discovery of their tandem versions in other species would bring biological relevance to these dimers. Indeed, with ongoing gene duplication, horizonal transfer, and phage-mediated genome recombination events, it is conceivable that singlet riboswitches can and have evolved into tandem versions to leverage cooperativity or other traits to effect a modified regulatory behavior, mirroring how certain tandem glycine riboswitches have devolved into singlets.

## 6. Dimeric Fluorogenic RNA Aptamers

Another notable class of RNAs that bind small molecules are the in vitro selected fluorescence-enhancing RNA aptamers [175]. When a weakly fluorescent small molecule binds to the aptamer, its intrinsic fluorescence can be dramatically enhanced making it a versatile tool to visualize RNA in cells. Different flavors of fluorescent RNAs have been designed or selected and now span the entire visible spectrum region [176]. Recently, the Corn RNA was selected to bind DFHO (3,5-difluoro-4-hydroxybenzylidene- imidazolinone-2-oxime), a compound that mimics the fluorescent properties of the red fluorescent protein (RFP). The Corn aptamer was shown to dimerize in solution through direct stacking between the two G-quadruplex base planes from each protomer in the absence of DFHO [177,178] (Figure 6). Although stacking of G-quadruplexes in solution is well-known [179], this structure represents the first example of a homodimeric RNA relying solely on stacking by G-quadruplexes and unpaired purines with no inter-protomer base pairing in the dimerization interface. The fluorophore binding pocket lies between both G-quadruplexes at the dimerization interface and a single DFHO molecule intercalates between the quartets through extensive stacking. Both G-quadruplexes and an auxiliary enclosure of 3 single-stranded adenosines collaborate to encapsulate DFHO, restrict its conformational freedom and augment its fluorescence. G-quadruplexes are a prominent motif that is frequently used to bind fluorescence dyes by these aptamers [175]. Interestingly, several new aptamers have been reported not to require this motif, such as the dimethylindole red (DIR) aptamers [180] and the dimeric fluorogenic RNA aptamer o-Coral, which is a RNA dimer selected to bind a dimeric, self-quenched fluorophore (sulforhodamine B) [181].

## 7. RNA Multimerization in Disease and Immunity

Dimerization is an effective and efficient way to explore sequence and structural space in constructing functional macromolecules, as a single mutation impacts both protomers at two spatially separate locations. Due to the reciprocity of inter-protomer contacts, dimeric interfaces can be readily stabilized, providing an expansive and stable platform to evolve diverse functions and regulatory circuits through cooperativity or allostery [1,4]. For instance, the proto-ribosome is believed to have evolved as a dimer [182,183]. From that perspective, it is perhaps surprising that compared to an abundance of protein dimers and multimers, RNA homodimerization seems to be a relatively rare biological phenomenon with only a handful of known examples. While it is all but certain that many more examples of RNA multimerization await discovery, there may be specific evolutionary pressures that have selected against them, namely their association with viral invasion and other types of stresses.

One primary limiting factor in the evolution of large RNA structures could have been the surveillance by the immune system. Nucleic acid multimerization often triggers innate immune responses [184,185,186,187]. In mammals, viral dsRNA is recognized by several families of proteins involved in innate immunity, including RIG-I, MDA5, TLR3, LGP2, PKR, etc. [188]. For instance, dsRNA-activated protein kinase R (PKR) is typically activated by long stretches of dsRNA >30 bp [189,190]. While monomeric HIV-1 TAR RNA or hepatitis delta virus (HDV) ribozyme do not activate PKR, their respective homodimers are strong PKR activators [191,192]. Indeed, homodimerization of these viral RNAs generally doubles the length of dsRNA regions providing a platform for PKR activation. Many RNA viruses, including HIV-1, employ a variety of strategies to escape such immune surveillance of their long dimeric gRNAs [193]. In another example, a single mutation A14G in the human mitochondrial tRNA^Leu^(UUR) creates a six nucleotide palindromic sequence in the D-stem-loop [194,195]. As a consequence, this mutant tRNA misfolds into a pathogenic homodimer with reduced aminoacylation efficiency leading to mitochondrial diseases, as well as PKR activation [196]. Furthermore, bidirectional transcription of the circular mitochondrial genome produces a heavy strand and a light strand that can hybridize to form dimeric mitochondrial dsRNA (mtdsRNA). Recently, mtdsRNA was found to trigger antiviral signaling, including MDA5 and PKR activation [197,198]. MDA5 forms filaments on long dsRNA >500 bp to trigger an interferon response [199]. As a result, PKR and MDA5 indirectly act as sentinels of mitochondrial integrity.

In addition to antiviral immunity, it is becoming increasingly clear that RNA multimerization is associated with general cellular stress [15,18]. Translation inhibition following stress leads to the formation of stress granules [200,201,202]. These higher-order ribonucleoprotein assemblies are formed through protein-protein, protein-RNA and RNA-RNA intermolecular interactions. Remarkably, many RNAs implicated in stress granules can form protein-free oligomers in vitro with unusual biophysical properties [18,203,204]. Understanding the molecular details or motifs underlying such RNA self-assemblies are an active area of investigation. Another instance of RNA oligomerization is exemplified by the trinucleotide repeat expansions (TNRE) disorders, a type of severe neurological diseases [16,205,206]. Pathologies occur only after a specific length threshold is reached [207]. Sufficiently long TNRE can form foci in vivo through phase separation [16]. This is largely mediated by intermolecular RNA-RNA interactions which drive condensation and aggregation. Structural work on shorter TNRE fragments revealed that such trinucleotide repeats can self-pair forming dsRNA-like structures via noncanonical base pairs interspersed between their canonically paired neighbors [206].

Taken together, it seems that RNA multimerization frequently leads to undesired cellular toxicity or immune responses. These factors could have restricted the evolution and adoption of RNA dimers and higher-order assemblies, at least within the cellular compartments wherein there is active immune surveillance.

## 8. Conclusions

In contrast to the abundance of protein dimers and multimers, there are only a handful of known cases of physiologically relevant RNA homodimers or oligomers. Nonetheless, in the laboratory, RNA multimerization is commonly observed both in solution and in crystallo. Our survey of representative multimeric RNA structures highlights the innate potential and inherent tendency of flexible RNA elements to exchange strands and swap domains to dimerize or multimerize. RNA appears to employ the same sort of interfaces and structural motifs that mediate tertiary structure formation to drive their higher-order assemblies. Occasionally, the *cis* interactions leading to monomer formation and the corresponding *trans* interactions leading to dimer formation compete with each other. The resulting monomer-dimer distribution can be in a dynamic equilibrium or static and non-interconverting. In both cases, the distribution is modulated by internal parameters, such as RNA concentration and thermal and folding history, as well as environmental parameters, including temperature, pH, ionic strength, divalent cations, crowding agents, osmolarity, etc. [208,209,210,211,212]. Thus, crystallized RNAs represent molecular entities or assemblies that do exist in solution and can also pack into ordered crystalline lattices.

How do RNA multimers form? Our comparative analyses show that long-range kissing-loop interactions and the exchange of complementary strands near RNA termini (e.g., frayed P1 regions) mediate most instances of dimerization. Nucleation of RNA protomers during the first encounter is typically initiated by the hybridization of short, consecutive segments of complementary sequences. This is akin to codon-anticodon interactions that occur on the ribosome and T-box riboswitches. Just like the codon-anticodon interactions, initial, sparse base-pairing interactions are usually not sufficiently stable and need to be further stabilized, either by extension of the base-pairing region (e.g., isomerization into extended dimers, such as the HIV DIS), or by flanking contacts. These flanking contacts can take the form of cross-strand stacking by adjacent, single-stranded, helix-capping purines (as in the case of codon-anticodon interactions), or coaxial stacking and concatenation of contiguous helices to boost overall stability (e.g., Guanidine-II and Glutamine-II riboswitches). Interestingly, these hybridization interactions can be split into two classes: those employing palindromic sequences and those that do not. While palindromic sequences are logically suited to drive dimer formation, non-palindromic but complementary sequences are more versatile. They have been effective in assembling variable numbers of multimers in head-to-tail processions, as seen in the *bcd* mRNA and ø29 pRNA. One exception to the hybridization-centric dimerization motif are RNAs that contain surface-exposed G-quadruplexes such as Corn. The strength of the stacking interactions between these large flat platforms alone is apparently sufficient to drive dimerization without base pairing. It is likely that future investigations will uncover naturally occurring G-quadruplex dimers or multimers in either cellular RNA or DNA. 

We anticipate that more biologically relevant RNA dimers and multimers will be discovered with the advent of novel crosslinking and RNA-seq-based methods to map RNA-RNA interactions genome wide in vivo [17,213,214,215]. For instance, transfer RNA fragments (tRFs) were recently shown to form homodimers that increase their stability against nucleases [216], while other G-rich tRFs form tetramers by forming G-quadruplexes [217]. The rise of cryo-electron microscopy (cryo-EM) will undoubtably expand the structural knowledge of RNA multimers alongside other methods in an integrative approach [218,219,220,221,222]. These experimental advances, in conjunction with improved molecular dynamics simulations and RNA structure prediction algorithms aided by artificial intelligence, will set the stage for better predicting the propensity of a given RNA sequence to oligomerize. 

Extensive RNA oligomerization or condensation is often associated with cellular stress or immune responses and is normally avoided in the cytosol of healthy cells. The interplay among soluble RNA oligomers and clusters, RNA and RNP condensates, and RNases, helicases, and chaperones is an area of intensive investigation, which is expected to transform our frame of thinking in the coming decade. Such mechanistic understanding will inform the design of molecules that modulate RNA oligomerization and open novel therapeutic avenues.

## Figures and Tables

**Figure 1 molecules-25-02881-f001:**
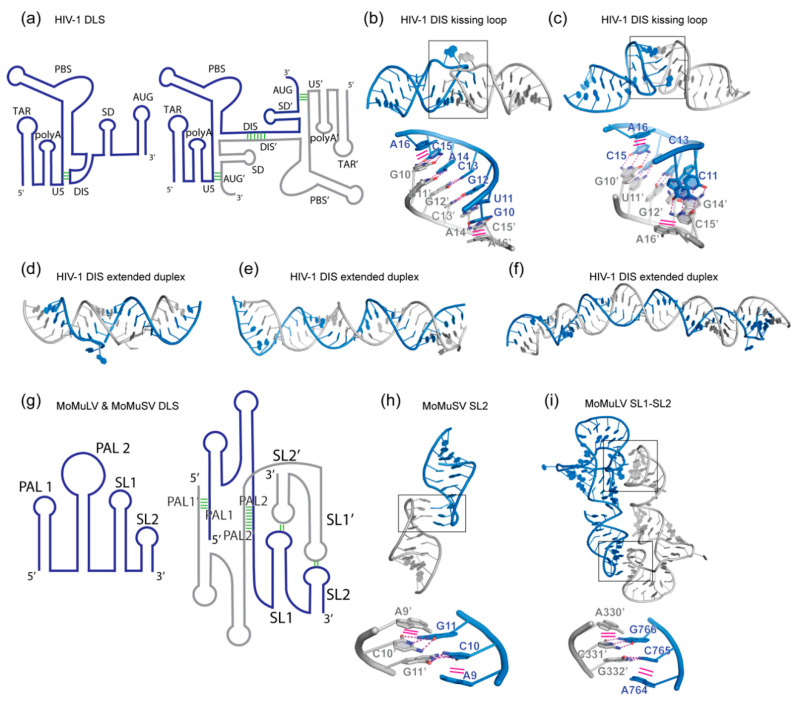
Homodimerization of retroviral genomic RNAs (gRNA). (**a**) Schematic view of the 5′ leader dimer linkage site (DLS) of human immunodeficiency virus type 1 (HIV-1) driving gRNA dimerization. Left: monomeric state with the dimerization initiation site (DIS) sequestrated by the U5 segment. Right: dimeric state driven by intermolecular DIS-DIS’ interaction. The *Gag* start site AUG’ element pairs in trans with the U5 segment. These conformational changes allow the HIV-1 genome to switch between translation (monomeric) and packaging (dimeric) functions. Structural elements followed by a prime symbol indicate that they belong to the other protomer. (**b**,**c**) Crystal and NMR structure of the DIS kissing loop interaction, respectively (PDB 2B8S and 1BAU). (**d**,**e**,**f**) Crystal, NMR, and Cryo-EM structures of the DIS homodimer in its extended duplex state (PDB 1Y99, 2GM0 and 6BG9). (**g**) Schematic view of the 5′ leader structure of the Moloney murine leukemia virus (MoMuLV) and Moloney murine sarcoma virus (MoMuSV) regulating gRNA dimerization. Four elements in this RNA drive its dimerization: palindromic sequences in PAL1 and PAL2 and kissing loop interactions between Stem-loop 1 (SL1) and Stem-loop 2 (SL2). (**h**) Structure of MoMuLV homodimer SL2 (PDB 1F5U). (**i**) Structure of MoMuSV homodimer SL1-SL2 (PDB 2L1F). Interface regions are boxed and detailed below each structure. Hydrogen bonding interactions are shown as magenta dashed lines, while stacking interactions are shown as solid magenta lines.

**Figure 2 molecules-25-02881-f002:**
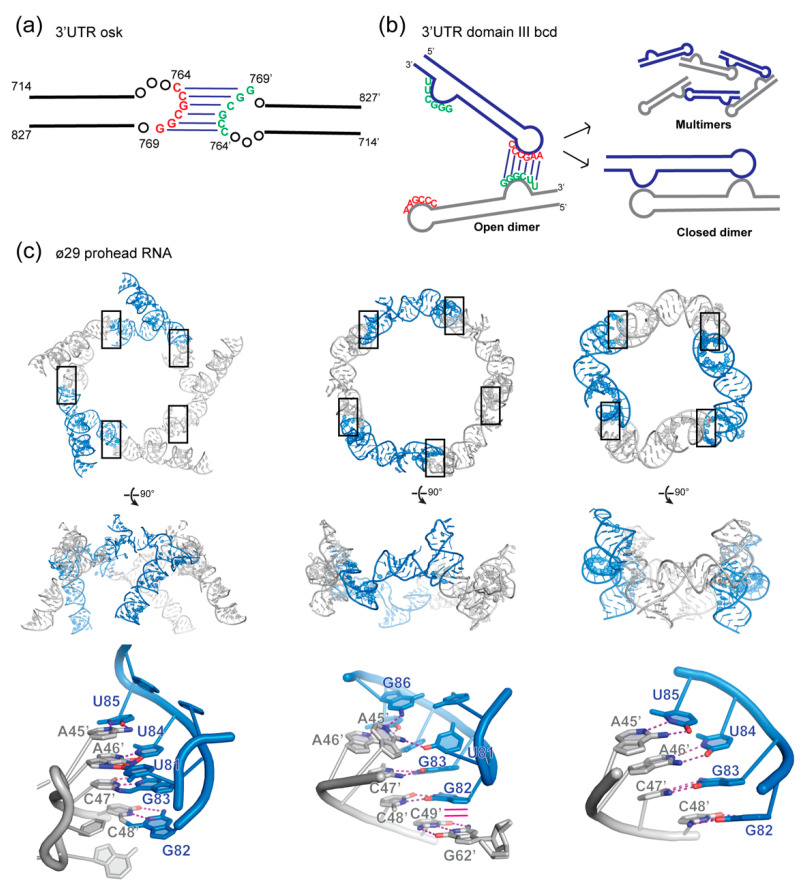
Homodimerization in mRNA transport. (**a**) *oskar* mRNA dimerize through a conserved GC rich kissing loop in the 3′-untranslated region (UTR). (**b**) *bicoid* mRNA dimerization motif relies on the complementarity of the apical loop and an internal bulge in the domain III of its 3′-UTR. This leads to formation of closed dimers or open dimers that can further oligomerize. (**c**) Structures of the prohead RNA of bacteriophage ø29 driven by kissing interactions between two stem-loops. From left to right, two cryo-EM structures of the pentameric assembly of pRNA (PDB 1FOQ and 6QYZ) and a crystal structure of a pRNA tetramer (PDB 3R4F). The conserved feature amongst the three structures is the Watson-Crick pairing between the 45AACC48 and 82GGUU85 segments, which drives higher-order assembly. Interface regions are boxed and detailed below each structure. Hydrogen bonding interactions are shown as magenta dashed lines, while stacking interactions are shown as solid magenta lines.

**Figure 3 molecules-25-02881-f003:**
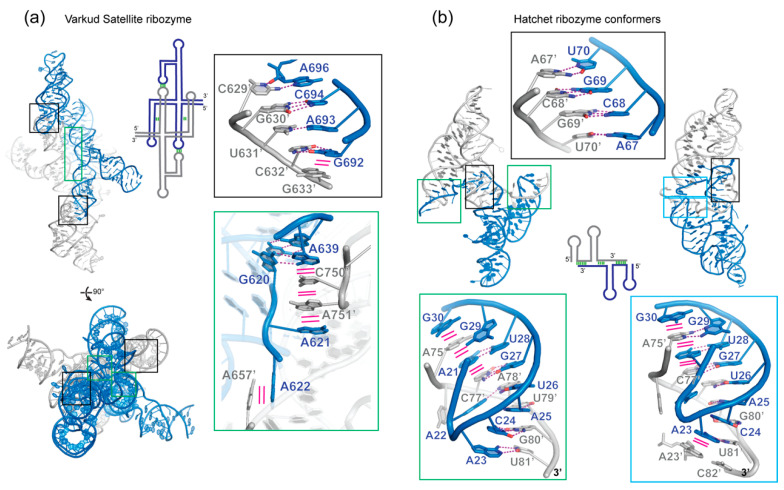
Structures of dimeric ribozymes. (**a**) Crystal structure of Varkud Satellite ribozyme (PDB code 4R4P & 4R4V) with the insets showing the intermolecular kissing loop interaction between stem-loop 1 (P1) and stem-loop 5′ (P5′), as well as the intercalation of stacked nucleobases between P1 and P6′. (**b**) Crystal structures of the Hatchet ribozyme (PDB 6QJ6 & 6QJ5) showing two types of dimeric interfaces. The top inset highlights the palindromic sequence paired with the same sequence of the second protomer. Hydrogen bonding interactions are shown as magenta dashed lines, while stacking interactions are shown as solid magenta lines.

**Figure 4 molecules-25-02881-f004:**
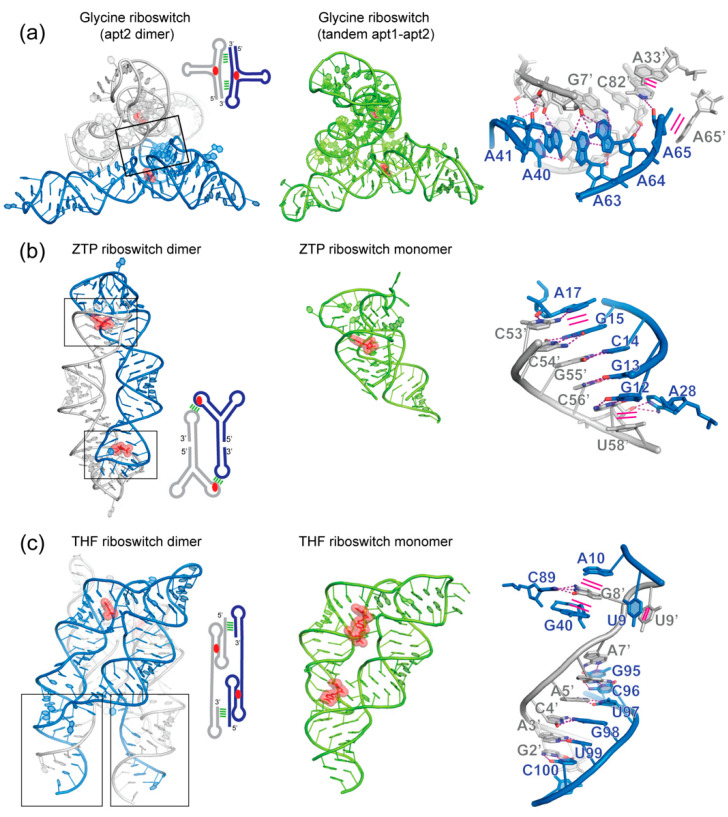
Comparison of monomeric and dimeric structures of the same riboswitches. (**a**) Glycine aptamer 2 crystallized in a dimer (PDB 3OWI) configuration similar to the tandem glycine aptamer1-aptamer2 (PDB 3P49). (**b**) ZTP riboswitch dimer structure (PDB 4XWF) juxtaposed with its monomeric version (PDB 5BTP). (**c**) THF riboswitch crystallized in dimeric (PDB 3SUY and 3SUX) and monomeric version (PDB 4LVV). Ligands are shown in stick representation and semi-transparent red spheres. Interface regions are boxed and detailed on the right of each dimeric structure. Hydrogen bonding interactions are shown as magenta dashed lines, and stacking interactions are shown as solid magenta lines.

**Figure 5 molecules-25-02881-f005:**
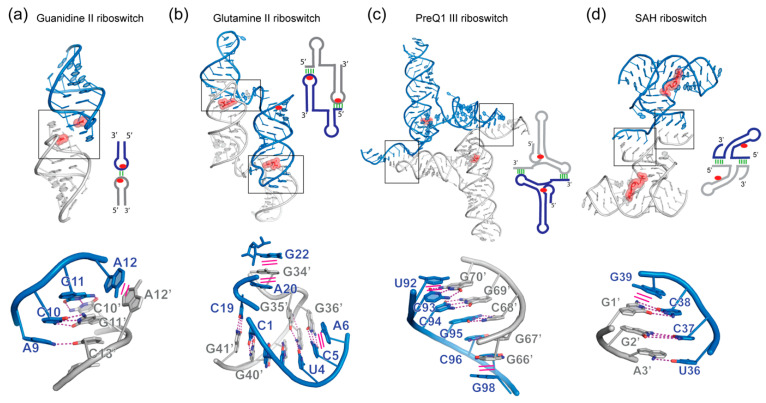
Additional in crystallo dimeric riboswitches. (**a**) Guanidine II riboswitch (PDB 5NDI and 5VJ9). (**b**) Glutamine riboswitch II (PDB 6QN3). (**c**) Class III preQ1 riboswitch (PDB 4RZD). (**d**) SAH riboswitch (PDB 3NPQ). Ligands are shown in stick representation and semi-transparent red spheres. Interface regions are boxed and detailed below each structure. Hydrogen bonding interactions are shown as magenta dashed lines, and stacking interactions are shown as solid magenta lines.

**Figure 6 molecules-25-02881-f006:**
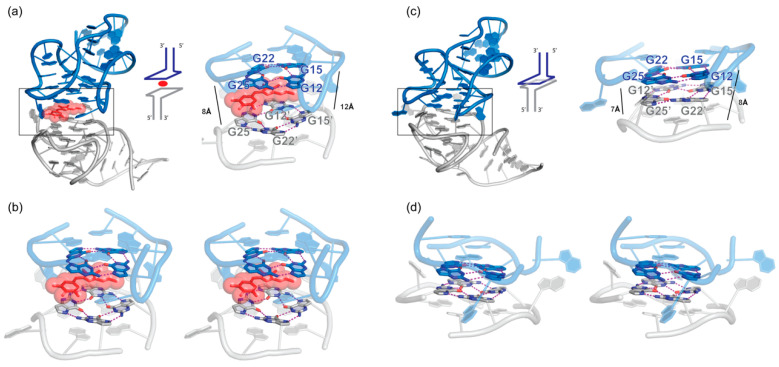
Dimeric structure of fluorogenic Corn RNA. (**a**) Bound to the DHFO (3,5-difluoro-4-hydroxybenzylidene- imidazolinone-2-oxime) ligand (PDB 5BJO) with a stereo view of its dimeric interface in (**b**). (**c**) In its apo form (PDB 6E80) with a stereo view of the collapsed dimeric interface in the absence of ligand (**d**). Ligands are shown in stick representation and semi-transparent red spheres. Interface regions are boxed and detailed below each structure. Hydrogen bonding interactions are shown as magenta dashed lines, and stacking interactions are shown as solid magenta lines.

**Table 1 molecules-25-02881-t001:** RNA homodimers discussed in this review. N.D.: not determined. The dimerization interface area (in Å^2^) was computed using PISA [69].

Name	PDB Code	Dimerization Motif	Interface (Å^2^)	Number of Nucleotides Per protomer	Method of Structural Determination
Human immunodeficiency virus type-1 DIS kissing loop	2B8S	Palindromic kissing loop (six nucleotides)	365.2	23	X-ray diffraction
	1BAU	Palindromic kissing loop (six nucleotides)	593.6	23	NMR
Human immunodeficiency virus type-1 DIS extended duplex	6BG9	Strand swapping	2251.6	47	Cryo-EM
	1Y99	Strand swapping	1083.1	23	X-ray diffraction
	2GM0	Strand swapping	1771.4	35	NMR
Moloney murine sarcoma virus SL2	1F5U	Palindromic kissing loop (two nucleotides)	246.7	18	NMR
Moloney murine leukemia virus SL1-SL2	2L1F	Palindromic kissing loop (two nucleotides)	1146.3	66	NMR
Oskar 3′-UTR	N.D.	Palindromic kissing loop (six nucleotides)	N.D.	N.D.	N.D.
Bicoid 3′-UTR	N.D.	Kissing loop (six nucleotides)	N.D.	N.D.	N.D.
Varkud satellite ribozyme	4R4P	Kissing loop (four nucleotides) Reciprocal base intercalation	2612.6	186	X-ray diffraction
	4R4V				X-ray diffraction
Hatchet ribozyme	6JQ5	Palindromic kissing loop (four nucleotides) Strand swapping	2157.3	82	X-ray diffraction
	6JQ6		2254.1	81	X-ray diffraction
Glycine riboswitch	3OWI	A-minor interactions Reciprocal base intercalation	1219	88	X-ray diffraction
	3OX0		1243	88	X-ray diffraction
Guanidine II riboswitch	5NDI	Palindromic kissing loop (two nucleotides) Reciprocal base intercalation	339.1	20	X-ray diffraction
	5VJ9		342.5	16	X-ray diffraction
Glutamine II riboswitch	6QN3	Pseudoknot Reciprocal base intercalation Base triple	1615.7	50	X-ray diffraction
preQ1 III riboswitch	4RZD	Pseudoknot	1212.6	101	X-ray diffraction
SAH riboswitch	3NPQ	Strand swapping	513	54	X-ray diffraction
ZTP riboswitch	4XWF	Pseudoknot	1945	64	X-ray diffraction
THF riboswitch	3SUX	Strand swapping	1860.9	101	X-ray diffraction
	3SUY		1909.6	101	X-ray diffraction
Corn RNA	5BJO	G-quadruplex stacking	569.7	36	X-ray diffraction
	6E80		670.2	36	X-ray diffraction
Internal ribosomal entry site (IRES) of Dicistroviridae	2IL9	Domain swapping	2087.5	142	X-ray diffraction
Designed GAAA tetraloop receptor complex	2I7Z	Tetraloop – tetraloop receptor	766.7	43	NMR

Abbreviations in the table: S-adenosylhomocysteine (SAH), 5-aminoimidazole-4-carboxamideriboside 5′-triphosphate (ZTP) and tetrahydrofolate (THF).

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
