# Peer review of "Structural Insights into RNA Dimerization: Motifs, Interfaces and Functions"

_molecules, 2020, doi:10.3390/molecules25122881_

Round 1

Reviewer 1 Report

The manuscript “Structural insights into RNA dimerization: motifs,  interfaces and functions” by Charles Bou-Nader and Jinwei Zhang is a nice well written review of various RNA dimerization motifs. The authors illustrate the importance of these motifs in regards to RNA structure formation, function and allude to their potential as therapeutic targets. Since these types of structures are key elements in RNA folding and assembly a review of the topic is important and beneficial. However, the authors should modify their manuscript in the following ways:

  1. The early use of the phrase palindromic sequences in the abstract and introduction of the paper appears to be too specific – self-annealing can also occur with complementary sequences, not just palindromic sequences. This is mentioned, but only later on.
  2. In section 4.2 there is a description of the hatchet ribozyme, please indicate in what RNAs they are found.
  3. Since this is a review paper it might be helpful to explain terms like π-π stacking in nucleobase interdigitation.
  4. The authors should cite several other authored papers such Lenotis/Westhof, Jaeger and Shapiro when discussing nanoparticle constructs formed by dimerization motifs.
  5. Single-stranded purines frequently cross-strand stack – Again, since this is a review paper it would be worthwhile to explain the importance of stacking – Petrov, Zirbel, Leontis, have a paper in RNA which explains the importance of stacking interactions.
  6. In the section on riboswitches, the authors should try to clarify more which structures might be artifacts of crystallization and which are natural forming. In addition, it would be helpful if the authors could expand somewhat on the function of the dimers that form. Do they form before ligand binding or are they present before binding. In figures 4 and 5 it would also be helpful if the ligands were shown in all of the 3D depictions as well as the 2D.
  7. In some of the blown up/insert structures it is very hard to decipher the interactions. It would be helpful if at least some were rendered in stereo, especially the more complicated ones such as figure 6.
  8. In general there should be some more discussion on the importance of Mg in the formation and function of these structures.

Author Response

The manuscript “Structural insights into RNA dimerization: motifs,  interfaces and functions” by Charles Bou-Nader and Jinwei Zhang is a nice well written review of various RNA dimerization motifs. The authors illustrate the importance of these motifs in regards to RNA structure formation, function and allude to their potential as therapeutic targets. Since these types of structures are key elements in RNA folding and assembly a review of the topic is important and beneficial. However, the authors should modify their manuscript in the following ways:

We thank the reviewer for his/her positive comments and constructive suggestions. We have modified the manuscript following the reviewer’s suggestions as detailed below.

  1. The early use of the phrase palindromic sequences in the abstract and introduction of the paper appears to be too specific – self-annealing can also occur with complementary sequences, not just palindromic sequences. This is mentioned, but only later on.

We agree with the reviewer that self-annealing occurs either with palindromic or complementary sequences. This has been corrected in the abstract and is further emphasized earlier in the introduction where we have added the following statement

  “The simplest, most common form of RNA-RNA interactions occur through base pairing of complementary single-stranded sequences, forming heterodimeric double-stranded RNA (dsRNA). These specific interactions are widespread in diverse forms of post-transcriptional gene regulation and RNA modification and processing such as the pairing of small interfering RNA (siRNA), microRNA (miRNA), or small nucleolar RNAs (snoRNAs) to their complementary RNA targets [8-12]..”

        2. In section 4.2 there is a description of the hatchet ribozyme, please indicate in what RNAs they are found.

The Hatchet ribozymes were identified in over 300 distinct genes from Veillonella sp. that often code for proteins with unknown functions (Weinberg Z. et al., Nat Chem Biol, 2015, 11(8), 606-10). We have clarified this in the manuscript by specifying that the Hatchet ribozyme was “discovered in several genes from Veillonella sp”.

            3. Since this is a review paper it might be helpful to explain terms like π-π stacking in nucleobase interdigitation.

In the introduction, we have now added a new paragraph that discussed the 3 types of forces that mediate RNA dimer formation: base pairing, base stacking, and tertiary interactions, which now include a definition and discussion for aromatic stacking. We now state:

These non-covalent, attractive interactions between polarized aromatic nucleobases daisy-chain the base pairs into helices and frequently further concatenate adjacent helices to form long, coaxial helical stacks. In so doing, aromatic-aromatic (or pi-pi) interactions to a large extent dictate the overall shape of most RNA tertiary and higher-order structures.”

          4. The authors should cite several other authored papers such Lenotis/Westhof, Jaeger and Shapiro when discussing nanoparticle constructs formed by dimerization motifs.

We have added the following references as suggested:

Afonin K. A. et al., Nat Nanotechnol, 2010, 5, 676-82

Grabow, W. W. et al., Nano Lett, 2011, 11, 878-87

Bindewald E. et al., ACS Nano, 2011, 5, 9542-51

Jaeger, L. E. el al., NAR, 2001, 29, 455-63

  1. Single-stranded purines frequently cross-strand stack – Again, since this is a review paper it would be worthwhile to explain the importance of stacking – Petrov, Zirbel, Leontis, have a paper in RNA which explains the importance of stacking interactions.

To emphasize the general importance of stacking in RNA structure and interactions underappreciated by many, we have added several sentences (also see point #3) in the 2nd paragraph of the introduction. We have also added this citation. To further the discussion of the ssRNA purines, we have added a statement near the end of section 2. We now state:

The stacking of such bases contributes significantly to the stability of the dimeric interface by minimizing exposure of hydrophobic bases to the solvent and by favorable enthalpy changes through the aromatic interactions.”

          6. In the section on riboswitches, the authors should try to clarify more which structures might be artifacts of crystallization and which are natural forming. In addition, it would be helpful if the authors could expand somewhat on the function of the dimers that form. Do they form before ligand binding or are they present before binding. In figures 4 and 5 it would also be helpful if the ligands were shown in all of the 3D depictions as well as the 2D.

To clarify this, at the beginning of the riboswitch section, we have added an overview and introductory paragraph. We now state:

In the following section, we discuss 7 classes of riboswitches that were crystallized as dimers. Among them, 3 classes (Glycine, ZTP, and THF) also crystallized as monomers, allowing direct comparisons with their dimeric counterparts. The biological relevance and potential functions of these dimers will also be discussed. In cases where the tandem aptamers are highly homologous, such as the glycine and guanidine-II riboswitches, dimeric structures of single aptamers in crystallo likely represented the way the natural tandem aptamers interact with each other and act in concert.”

For several classes we already discuss the biological relevance of the dimers (glycine, ZTP, etc). For others we have now added explicit discussions at the end of their sections. Such as:

Although these domain-swapped dimer configurations presumably do not occur in vivo, the dimeric structures nonetheless were able to be interpreted to understand the functional monomers.”

Importantly, nearly identical ZTP-binding pockets were observed in the two structures, suggesting that the crystallographic dimer structure from S. odontolytica actually captured the biological relevant RNA-ligand interface.”

Similarly, this dimerization interface is not thought to function in vivo.”

We now expand a bit on the dimeric functions at the end of the riboswitch section. We comment that the functional assignments could change in the future, as more variants are discovered. For this reason, we refrain from classifying them as artifacts. We now state:

For riboswitches that are only known to function as singlet monomers but crystallized as dimers, it remains possible that future discovery of their tandem versions in other species would bring biological relevance to these dimers. Indeed, with ongoing gene duplication, horizonal transfer, and phage-mediated genome recombination events, it is conceivable that singlet riboswitches can and have evolved into tandem versions to leverage cooperativity or other traits to effect a modified regulatory behavior, mirroring how certain tandem glycine riboswitches have devolved into singlets.”

We have added the ligands to all 2D depictions in figures 4 and 5 as suggested by the reviewer. For clarity of the blown up/insert structures we do not depict the ligands in order to more clearly illustrate the RNA-RNA intermolecular interactions, as raised in point 7.

       7. In some of the blown up/insert structures it is very hard to decipher the interactions. It would be helpful if at least some were rendered in stereo,  especially the more complicated ones such as figure 6.

We have added a stereo view for the dimerization interface in figure 6.

      8. In general there should be some more discussion on the importance of Mg in the formation and function of these structures.

Indeed, Mg2+ is required for proper folding, tertiary structure formation and stabilization of RNA structure as well as for catalysis by some ribozymes. We have added a general discussion in the second paragraph of the introduction. We state:

These tertiary and quaternary interactions between discrete RNA elements can be dramatically stabilized by the presence of Mg2+ ions and to a lesser extent by monovalent cations such as K+. Mg2+ ions play the dual roles of serving as diffuse counter ions that ameliorate the electrostatic stress from juxtaposing densely charged phosphate backbones, as well as bridging specific tertiary contacts in the form of chelated ions.”

We are unable to discuss this in further detail due to the lack of specific studies that rigorously addresed the effect of Mg2+ on the formation of RNA homodimeric structures.

Reviewer 2 Report

The authors present a very through and detailed analysis of RNA dimerization.  The article is dominated by an extensive set of examples of RNA dimers with both structural and functional details given.  While the authors present an encyclopedic description of various RNA dimers, the article provides few clear insights into the phenomenon.  It would benefit the readability enormously for the authors to organize their article around major themes rather than simple RNA classes and make more clear what the themes are from the outset of the article.  This reviewer suggests the authors consider the following comments in a revised manuscript:

page 1, lines 35-39. Comparison of RNA and proteins with respect to the nature of homo-oligomeric states is an “apples and oranges” comparison due to the very nature of the macromolecules. The simplest way for nucleic acids to associate with one another is through Watson-Crick pairing, leading to sense/antisense interactions, which is common in RNA biology and, unless the sequence is palindromic, is obligatorily heterodimeric. Conversely, proteins typically interact via tertiary structures in which the monomers or higher order complexes can fold into a three-dimensional shape that are self-complimentary. It is notable that in both cases, often the reason for multimerization is regulatory, however. Indeed multimerization is prevalent in both protein and RNA biology.

page 2, lines 44-45. Again, a difficult comparison is drawn between the biological homodimers of the first paragraph and the in vitro artifacts of the second. In the former, the authors cite the PTC as a pseudo-dimer that is formed through tertiary structure while the in vitro dimers are generally through Watson-Crick pairing of limited regions of the RNA.  Would the in vitro dimers be more comparative to the frequent use of sense/antisense interactions in biology?

page 4, Table 1. Can the authors comment on how accurate PISA is for RNA-RNA interactions? Further, these numbers are not discussed in the text, so it is unclear what to think of them.  Also, the surface areas are sparsely mentioned, and so how these calculations advance our understanding of RNA-RNA interactions is again unclear.

pages 4 – 15. The authors present a very long list of examples of homodimeric RNAs, both natural and in vitro. This listing is, frankly, a tedious read for several reasons.  First, the treatment of these various RNAs is often unbalanced, with some RNAs detailed in great detail with regards to their biology, structure and function.  Conversely, some RNAs are treated with only a few sentences and it is not clear why these RNAs are worth description beyond their inclusion in their comprehensive list (Table 1).  It would help the article significantly to choose a subset of RNAs that best exemplify the themes that the authors wish to convey to the reader.  Second, the authors make little attempt to organize their descriptions according to structural themes.  Instead, the RNAs are organized according to their biology (viral, mRNA, ribozyme, riboswitch, etc…).  This organization does nothing to help emphasize themes of homodimerization.  In the end, the reader is left with very little significant insights into this issue.  While the conclusions begin to make these connections, a reorganization of the main text would greatly assist in the reader in developing a clear sense of (1) how RNAs homodimerize and (2) how this homodimerization facilitates function.  

Author Response

The authors present a very through and detailed analysis of RNA dimerization.  The article is dominated by an extensive set of examples of RNA dimers with both structural and functional details given.  While the authors present an encyclopedic description of various RNA dimers, the article provides few clear insights into the phenomenon.  It would benefit the readability enormously for the authors to organize their article around major themes rather than simple RNA classes and make more clear what the themes are from the outset of the article.  This reviewer suggests the authors consider the following comments in a revised manuscript:

We thank the reviewer for his/her insightful comments and constructive suggestions. We agree that the descriptions can sound like a laundry list of known instances followed by some analysis. Therefore, we have attempted to reorganize the entire narrative under several themes or motifs rather than biological categories. However, we did not succeed in achieving improved clarity as the sections became more imbalanced, because (a) there are insufficient number of recognizable themes and motifs to classify the examples under; (b) certain motifs became so dominant (such as strand and domain swapping) and others almost insignificant. In addition, as mentioned in the text, we had intended for the review to be a comprehensive meta-analysis, or unbiased survey of most known instances of RNA dimerization, and in so doing to detect patterns and trends that would emerge. This approach, perhaps different from some other reviews, naturally led to this detail-heavy format that did not start from themes and motifs that were already known and declared at the beginning. In light of all this, we propose to maintain the current narrative organization and work to address other issues, as detailed below.

page 1, lines 35-39. Comparison of RNA and proteins with respect to the nature of homo-oligomeric states is an “apples and oranges” comparison due to the very nature of the macromolecules. The simplest way for nucleic acids to associate with one another is through Watson-Crick pairing, leading to sense/antisense interactions, which is common in RNA biology and, unless the sequence is palindromic, is obligatorily heterodimeric. Conversely, proteins typically interact via tertiary structures in which the monomers or higher order complexes can fold into a three-dimensional shape that are self-complimentary. It is notable that in both cases, often the reason for multimerization is regulatory, however. Indeed multimerization is prevalent in both protein and RNA biology.

We agree with the reviewer that all dsRNAs are indeed true dimers and abundant in cells. However, to focus the review on the RNA dimers that are mediated at least in part by tertiary structural contacts, we implicitly excluded the simple RNA dimers that form solely via hybridization. To clarify this, we have added to the second paragraph of the introduction the acknowledgements that simple dsRNA are indeed valid dimers but explicitly limit the focus of the review to those that involve some tertiary contacts. We state:

The simplest, most common form of RNA-RNA interactions occur through base pairing of complementary single-stranded sequences, forming heterodimeric double-stranded RNA (dsRNA). These specific interactions are widespread in diverse forms of post-transcriptional gene regulation and RNA modification and processing such as the pairing of small interfering RNA (siRNA), microRNA (miRNA), or small nucleolar RNAs (snoRNAs) to their complementary RNA targets, etc. [8-12]. These hybridization events produce heterodimeric dsRNAs that are abundant in cells. However, for the purpose of this review, we generally exclude simple dsRNAs that form solely through sense-antisense hybridization, and treat dsRNA as monomers, in order to focus on dimerization that involve tertiary structural contacts.”

page 2, lines 44-45. Again, a difficult comparison is drawn between the biological homodimers of the first paragraph and the in vitro artifacts of the second. In the former, the authors cite the PTC as a pseudo-dimer that is formed through tertiary structure while the in vitro dimers are generally through Watson-Crick pairing of limited regions of the RNA.  Would the in vitro dimers be more comparative to the frequent use of sense/antisense interactions in biology?

We acknowledge the valid point brought by the reviewer. As discussed in the preceding point, our point of observation might have been somewhat different from that of the reviewer, in that we emphasize on the tertiary and quaternary RNA-RNA interactions. Interestingly, we believe the in vitro misfolded multimers also involve higher order interactions, although few researchers are sufficiently motivated to investigate this. For instance, when we refold tRNA-Gly with certain thermal procedures that were effective for RNAse P refolding into monomers, we obtained dramatic ladders that include dimers, tetramers, 8-mer, 16-mer, etc. By the comparison we meant to draw the readers’ attention to the reported discrepancy between the in vivo and in vitro behaviors of the same RNA, and to suggest that the chemical nature of RNA in fact encourages multimerization.

To draw the comparison between in vitro dimers to the sense/antisense dsRNA, we added a discussion in the 4th paragraph of the introduction:

Most of such in vitro multimers are believed to occur via fortuitous pairing of short segments of complementary sequences, when the RNA was heat-denatured and cooled slowly —a process that encourages strand annealing. As such these in vitro multimers are similar in origin to those dsRNAs produced by sense-antisense hybridization events in cells.”

page 4, Table 1. Can the authors comment on how accurate PISA is for RNA-RNA interactions? Further, these numbers are not discussed in the text, so it is unclear what to think of them.  Also, the surface areas are sparsely mentioned, and so how these calculations advance our understanding of RNA-RNA interactions is again unclear.

As mentioned in our response to the first point, we had intended the article to be a comprehensive reference material for readers who might be interested in using the tabulated information for comparative purposes in their own work, and for an overview of the approximate range of the dimerization parameters.

In terms of the PISA calculations, the calculation of the areas of burial of solvent-accessible areas (ASA) is largely independent of the chemical nature of the macromolecules. We had compared PISA numbers to those derived from other software such as StructTools, which yielded comparable numbers. The accuracy of the energetics, however, is less clear. The predicted G0dissociation may not adequately and accurately estimate the free energy for complex interfaces between RNAs, especially the strength of stacking interactions. A case in point is that the G0dissociation of Corn RNA is quite positive whereas it dimerizes readily in solution, suggesting that the software is not able to accurately predict stacking energetics. In light of this we have removed the G0dissociation columns but retained the interfacial buried surface areas. The latter correlates well with the number of interfacial base-pairs and could be a useful guide for readers to select a natural homodimeric RNA as a scaffold to engineer RNA nanodevices. 

pages 4 – 15. The authors present a very long list of examples of homodimeric RNAs, both natural and in vitro. This listing is, frankly, a tedious read for several reasons.  First, the treatment of these various RNAs is often unbalanced, with some RNAs detailed in great detail with regards to their biology, structure and function.  Conversely, some RNAs are treated with only a few sentences and it is not clear why these RNAs are worth description beyond their inclusion in their comprehensive list (Table 1).  It would help the article significantly to choose a subset of RNAs that best exemplify the themes that the authors wish to convey to the reader.  Second, the authors make little attempt to organize their descriptions according to structural themes.  Instead, the RNAs are organized according to their biology (viral, mRNA, ribozyme, riboswitch, etc…).  This organization does nothing to help emphasize themes of homodimerization.  In the end, the reader is left with very little significant insights into this issue.  While the conclusions begin to make these connections, a reorganization of the main text would greatly assist in the reader in developing a clear sense of (1) how RNAs homodimerize and (2) how this homodimerization facilitates function.  

As mentioned in our response to the first point, our intention for the review was to serve as an unbiased “field survey” of sorts. As we did not have any pre-conceived conclusions, we try to avoid unintended bias by being thorough and non-selective in the examples that we can find, which are not too numerous to cover. An outcome of the survey was that certain themes are dominant, such as strand-swapped dimers. If we had organized the review (and we tried) based on structural themes, then 80-90% of the examined RNA would fall into this single category, making the review more imbalanced. On top of the strand-swapping, there are further elaborations such as cross-strand stacking, etc that are diverse and can be crucial for the dimerization. In addition, the organization based on biology has the potential utility of detecting trends and patterns that are correlated with the biological function.

The length and level of detail of the RNA categories largely correlated with the available literature on the RNA in question, in terms of coverage and depth. For instance, the glycine riboswitches have received significantly more attention and thus much more is known about the system, and we dedicated more space discussing them.

Round 2

Reviewer 2 Report

The authors have adequately addressed the comments made by this reviewer with respect to the first submission of this manuscript.  While I still believe that the organization of the manuscript makes this a bit of a "laundry list" read, I also appreciate the author's attempt to reorganize the paper.  I have no further concerns regarding this work.

Author Response

We thank the reviewer for his/her positive comments and appreciate the constructive exchange that has improved the manuscript.